# Screening for Antifungal Indigenous Lactobacilli Strains Isolated from Local Fermented Milk for Developing Bioprotective Fermentates and Coatings Based on Acid Whey Protein Concentrate for Fresh Cheese Quality Maintenance

**DOI:** 10.3390/microorganisms11030557

**Published:** 2023-02-22

**Authors:** Agne Vasiliauskaite, Justina Mileriene, Beatrice Kasparaviciene, Elvidas Aleksandrovas, Epp Songisepp, Ida Rud, Lars Axelsson, Sandra Muizniece-Brasava, Inga Ciprovica, Algimantas Paskevicius, Jurgita Aksomaitiene, Ausra Gabinaitiene, Dainius Uljanovas, Violeta Baliukoniene, Liis Lutter, Mindaugas Malakauskas, Loreta Serniene

**Affiliations:** 1Department of Food Safety and Quality, Veterinary Academy, Lithuanian University of Health Sciences, Tilzes Str. 18, LT-47181 Kaunas, Lithuania; 2BioCC OÜ, Riia 181A-233, 50411 Tartu, Estonia; 3Nofima, Norwegian Institute of Food, Fisheries and Aquaculture Research, P.O. Box 210, NO-1431 Ås, Norway; 4Faculty of Food Technology, Latvia University of Life Sciences and Technologies, Rigas Str. 22A, LV-3002 Jelgava, Latvia; 5Laboratory of Biodeterioration, Research Nature Research Centre, Akademijos Str. 2, LT-08412 Vilnius, Lithuania

**Keywords:** *Lactobacilli*, safety, antibiotic resistance, antifungal activity, acid whey protein concentrate, fermentate, edible coating, acid-curd cheese

## Abstract

The demand for healthy foods without artificial food additives is constantly increasing. Hence, natural food preservation methods using bioprotective cultures could be an alternative to chemical preservatives. Thus, the main purpose of this work was to screen the indigenous lactobacilli isolated from fermented cow milk for their safety and antifungal activity to select the safe strain with the strongest fungicidal properties for the development of bioprotective acid whey protein concentrate (AWPC) based fermentates and their coatings intended for fresh cheese quality maintenance. Therefore, 12 lactobacilli strains were isolated and identified from raw fermented cow milk as protective cultures. The safety of the stains was determined by applying antibiotic susceptibility, haemolytic and enzymatic evaluation. Only one strain, *Lacticaseibacillus paracasei* A11, met all safety requirements and demonstrated a broad spectrum of antifungal activity in vitro. The strain was cultivated in AWPC for 48 h and grew well (biomass yield 8 log_10_ cfu mL^−1^). *L. paracasei* A11 AWPC fermentate was used as a vehicle for protective culture in the development of pectin-AWPC-based edible coating. Both the fermentate and coating were tested for their antimicrobial properties on fresh acid-curd cheese. Coating with *L. paracasei* A11 strain reduced yeast and mould counts by 1.0–1.5 log_10_ cfu mL^−1^ (*p* ≤ 0.001) during cheese storage (14 days), simultaneously preserving its flavour and prolonging the shelf life for six days.

## 1. Introduction

Lithuanian acid-curd cheese is one of the most affordable and, thus, popular cheeses consumed in Baltics. Due to low salt and high moisture contents and post-processing contamination due to manual operations, this cheese is prone to mould and yeast spoilage resulting in a short shelf-life (7–8 days) [1]. Despite the use of preventive approaches during technological processes, rapid spoilage of this cheese leads to significant food waste and losses concerning cheese manufacturers and consumers [2]. On the other hand, Europe, responding to growing consumer demand for food with fewer chemical preservatives, puts pressure on the producers to increase the production of preservative-free products [3].

Bioprotective cultures are being considered as one of the possible measures against food spoilage with wild-type, indigenous LAB isolated from various local sources leading the latest trend [4,5]. Even though they are defined as food-grade bacterial strains and are ‘generally recognised as safe’ (GRAS) or have QPS (Quality Presumption of Safety) status [6,7], newly isolated indigenous strains intended for use in the food industry must pass strict screening processes related to their safety requirements. Antibiotic resistance, and haemolytic and enzymatic activity, among other safety properties, remain essential in the selection of strains for application in the food production [8].

It has been reported that the application of protective cultures requires a proper carrier to provide protection and support the survival of the strain on the cheese matrix [9]. Thus, the incorporation of such cultures in an edible coating—a thin edible membrane containing hydrocolloids (proteins and polysaccharides) and able to retain product freshness during storage has been reported to be an effective vehicle for living cells [10]. Hydrocolloids not only facilitate the incorporation of the strain but, throughout the extent of the latency phase [11], also enhance the effectiveness of inhibition of the growth of spoilage microflora, thus increasing the stability, safety, and shelf-life of the product [12]. To date, only a few studies have investigated the effectiveness of incorporating viable protective LAB in such coatings; neither of them was applied to the cheese matrix [13].

Recent studies show that for the development of the coatings intended for cheese applications, powdered sweet whey proteins [14,15], along with other powdered ingredients, are acquired using water as a base [16]. The production of such coating solutions may be costly. Therefore, a new trend of reusing dairy waste in its natural liquid form, such as sweet whey [1] and especially acid whey [17], is emerging. Since dairy factories most frequently use ultrafiltration to separate liquid acid whey protein concentrate (AWPC) from acid whey, this technology offers a solid foundation for coating formulations that contain proteins and sugars to support strain survival, water to dilute coating ingredients, and various postbiotics that may function as antimicrobial agents in food applications. Reintroducing AWPC back into food production brings along environmental and economic benefits as well [18].

On the other hand, AWPC may serve as an affordable LAB cultivation media, providing another alternative for acid whey repurposing. De Man, Rogosa, and Sharpe (MRS) broth are being used extensively for the growth of the LAB [19]. However, the high cost of this medium, and time-consuming biomass processing procedures, limit its use under specific laboratory conditions. So far, few publications have provided information on the use of acid whey in the preparation of LAB culture medium. Dudkiewicz et al. [20] investigated the filtration and sterilisation of acid whey to cultivate different yeast strains. There was also an attempt to grow LAB in an acid whey-based medium supplemented with low levels of yeast extract [21]. Employing AWPC for acid-tolerant LAB biomass production, thus utilising whey proteins, peptides, and naturally occurring sugars as an energy source for the strain growth instead of harvesting it from MRS broth, maybe a valid, natural, and cost-effective biomass growth method. Due to no colour or flavour additives, AWPC fermentates may be directly incorporated into coating solutions omitting growth media removal in LAB biomass production. These opportunities have not been yet investigated; therefore, the aim of our study was to screen the indigenous lactobacilli isolated from fermented cow milk for their safety and antifungal activity to select the safe strain with the strongest fungicidal properties for the development of bioprotective AWPC-based fermentates and their coatings intended for fresh cheese quality maintenance.

## 2. Materials and Methods

### 2.1. Materials

We isolated and identified 12 lactobacilli strains from local naturally fermented (37 °C, 96 h) Lithuanian cow milk (4.20% fat, 3.00% protein, 4.50% lactose, pH 6.60), which were stored at −80 °C in MRS broth (Oxoid, Basingstoke, UK) in the presence of 30% glycerol. Before experiments, strains were revitalised in MRS broth by growing for 48 h at 37 °C.

The target fungi strains used in this work were yeasts *Debaryomyces hansenii* (BTL, M-31/Deb. 4 (T)), *Yarrowia lipolytica* (BTL, M-48/C.6.1), *Candida glabrata* (BTL, 8B), *Candida albicans* (BTL, M-7/C.17) and moulds *Mucor racemosus* (BTL, G-134/1064), *Cladosporium herbarum* (BTL, G-86/KŠ-2/7), *Penicillium commune* ML 21-3, *Aspergillus versicolor* (CBS 113090) which were kept in the culture collection of Nature Research Centre, Institute of Botany, Vilnius, Lithuania. These strains were cultured in Sabouraud dextrose agar (SDA) (Oxoid, Basingstoke, UK) at 30 °C for 2 days for yeasts and at 25 °C for 5 days for moulds.

Two growth media for LAB cultivation—MRS broth and acid whey protein concentrate (AWPC; 0.5% fat, 2.18% protein, 9.72% lactose, pH 4.7), produced by ultrafiltration of acid whey at AB Kauno pienas, Lithuania) were chosen for the study.

Fresh acid-curd cheese was purchased from a nearby artisanal dairy factory (9.0% fat, 14.7% protein, 3.5% lactose, pH 4.63). It was prepared using traditional methods, including thermocoagulation of soured milk, which involved pasteurising standardised bovine milk (4.00% fat, 3.00% protein, 4.50% lactose, 8.10% non-fat solids, pH 6.60), cooling it to 28 °C, and adding the commercial starter. To create curd, soured milk was heated for 90 min to 50 ± 1 °C after casein coagulation (12 h). The acid-curd that was produced after the whey separation was kept in a refrigerator at a temperature of 4 ± 1 °C.

### 2.2. Identification of Lactic Acid Bacteria

Utilising a maximum recovery diluent (Oxoid, Basingstoke, UK), a ten-fold dilution series was performed on samples of fermented milk. Selected dilutions were applied in triplicates on MRS agar (Biolife, Milan, Italy) and then incubated in jars with Anaerogen (Oxoid, Basingstoke, UK) under anaerobic conditions for 48 h. After incubation, representative colonies from each plate were chosen and purified on MRS agar several times. The isolation was achieved on MRS agar by analysing morphological parameters (colony and cell morphology), as well as using biochemical assays (Gram staining and catalase test) [22]. Until further investigation, LAB characteristic colonies (Gram-positive, catalase-negative) were kept at 80 °C in MRS broth (Oxoid, Basingstoke, UK), which was supplemented with 30% glycerol.

### 2.3. Identification of Lactobacilli with MALDI-TOF MS Biotyper

Using the Matrix-assisted laser desorption/ionisation Time of flight (MAL-DI-TOF, Bruker, Germany) mass spectrometry method and the 3.0 software package (Bruker, Germany), measurements were made after the isolates were identified using biochemical assays. As per the procedure outlined by Dec et al. [23], the mass spectrometry was calibrated using Bruker’s bacterial test standard (Bruker Daltonics).

### 2.4. Safety Assessment

#### 2.4.1. Haemolytic Activity

The strain’s haemolytic activity was assessed on blood agar plates that contained 5% sheep blood (E & O Laboratories Ltd., Bonnybridge, UK). Haemolytic activity was measured after 48 h of incubation at 37 °C and recorded as β-haemolysis, α-haemolysis and γ-haemolysis represented as clear zones, green zones, or no haemolysis around the colonies [24].

#### 2.4.2. Antibiotic Susceptibility

Antibiotic susceptibility was evaluated on Mueller-Hinton agar (Oxoid, Basingstoke, UK) using MIC Test Strips (Liofilchem, Roseto degli Abruzzi, Italy) following the manufacturer’s instructions. Some of the strains were unable to grow on Mueller-Hinton agar. Therefore, the analysis was repeated using Mueller-Hinton agar enriched with 10% of MRS agar (Biolife, Milano, Italy). Chloramphenicol, clindamycin, streptomycin, gentamicin, tetracycline, erythromycin, ampicillin, vancomycin, and kanamycin were the antibiotics examined. Minimum Inhibitory Concentrations (MIC), which are reported in g mL^−1^, were calculated using the MIC reading scale. Isolates were designated as resistant and susceptible following the breakpoints given by European Food Safety Authority (2012) [25].

#### 2.4.3. Enzymatic Activities

For each of the chosen strains, the enzymatic activity of enzymes was assessed using the API ZYM kit (bioMerieux, Marcy-l’Étoile, France). The API ZYM strips experiment was carried out as previously described by Kondrotiene et al. [26], and the outcomes were assessed in accordance with the recommendations of the manufacturer. Based on the strength of colour production, changes in colour were rated from 0 to 5. Colour reaction grade 0 was interpreted to correspond to a negative reaction, grades 1 and 2 to a weak reaction (5 to <20 nmol substrate metabolised), and grades 3 through 5 to a strong reaction (>20 nmol substrate metabolised).

### 2.5. Antifungal Activity Assays of L. paracasei A11

After meeting all safety requirements, *L. paracasei* A11 was screened for antifungal activity in vitro. The antifungal activity of the strain was tested using a dual culture overlay assay as described by Magnusson et al. [27] with some modifications. Isolates were inoculated in two lines 2 cm long on MRS agar (Biolife, Milano, Italy) plates and were incubated at 37 °C for 48 h in jars using Anaerogen (Oxoid, Basingstoke, UK). Then the plates were overlaid with 5 mL SDA (Oxoid, Basingstoke, UK) containing 10^5^ mould spores or yeast cells per mL. The mould spores and yeast cells were collected in sterile deionised water, and the quantities of each were measured using a hemocytometer, and adjusted to 10^5^ spores/cells per mL [27]. The inhibitory zones were evaluated following a 48-h aerobic incubation period at 25 °C. Percentage grades were assigned based on the inhibition growth area per inoculation streak in relation to the overall area of petri dish inhibition. The scale that was employed was as follows: +++, no fungal growth on 8% of plate area/bacterial streak; ++, no fungal growth on 3–8% of plate area/bacterial streak; +, no fungal growth on 0.1–3% of plate area/bacterial streak; -, no apparent inhibition. Three duplicates of the inhibition tests were performed.

### 2.6. Cultivation of L. paracasei A11 and Coating Preparation

#### 2.6.1. Cultivation of *L. paracasei* A11 in AWPC

The *L. paracasei* A11 strain met all tested safety requirements and expressed strong broad-spectrum antifungal activity and was chosen for further cultivation in AWPC and preparation of bioactive edible coating. The cultivation of selected *L. paracasei* A11 was carried out in AWPC and MRS broth (Oxoid, Basingstoke, UK). The strain was simultaneously cultivated in the bioreactor and in the flasks in the thermostat. Cell growth kinetics of the strain was assessed using the bioreactor (RTS-1C, Biosan, Reverse-Spin technology) with software and the function of monitoring the growth of microorganisms in real-time following the manufacturer’s instructions. The strain was obtained after growing it on MRS agar at 37 °C for 48 h under anaerobic conditions. AWPC was pasteurised for 10 min at 93–95 °C in a water bath. 29 mL of cooled down to 37 °C MRS and AWPC were placed in separate 30 mL TPP TubeSpin vessels with a membrane filter (29 mL) and 1 mL *L. paracasei* A11 suspension in deionised water (adjusted by densitometer (Biosan, Latvia) to 1 McFarland unit) was added to each vessel. The tubes were placed in a bioreactor for 48 h at 37 °C under aerobic conditions. The cultivation’s optical density (Owas measured at 850 nm wavelength every 10 min with the agitation speed of 110 rpm, while the growth rate of the strain was calculated automatically.

To estimate the yield of biomass, the cultivation of the strain was carried out in a sterile 250 mL Erlenmeyer flask containing 29 mL of MRS and AWPC media and 1 mL of strain suspension in deionised water. The flasks were kept in the thermostat at 37 °C for 48 h under aerobic conditions. The experiment was performed in triplicate under the same conditions.

After the cultivation in the thermostat, AWPC fermentate was used for the application on the cheese and for the preparation of the bioprotective coating.

#### 2.6.2. Coating Preparation

Coating formulations were produced by Ramos et al. [14] with some modifications (Figure 1). The stock coating solution was made by dissolving the coating components (5% glycerol (*w*/*w*), 2% pectin (*w*/*w*), 0.2% Tween (*w*/*w*), and 2% sunflower oil (*w*/*w*)) in 50% of the AWPC amount required for in the recipe (90.8%). The solution was homogenised at 15,000 rpm for 3 min, pasteurised in a water bath for 10 min at 93 to 95 °C, and then cooled to 35 °C. The missing half of AWPC was used for the cultivation of *L. paracasei* A11. Two coatings—control (Coating) and experimental (Coating + A11) with incorporated *L. paracasei* A11—were prepared from the stock solution. The production of the control coating was finalised by mixing in the missing half of the required AWPC amount to the stock solution. The same amount of AWPC fermentate containing 8.3 log_10_ cfu mL^−1^ of *L. paracasei* A11 was added to the stock solution and thoroughly mixed to incorporate the strain into the experimental coating (Coating + A11) with a final concentration of 7.7 log_10_ cfu mL^−1^.

### 2.7. AWPC, AWPC Fermentate, and Coating Application on Acid-Curd Cheese

Immediately before applying the coating, the fresh acid-curd cheese (100 g) was distributed into plastic cups and lightly pressed. Plain pasteurised AWPC (C + AWPC), AWPC fermentate (AWPC Fermentate A11) and both coatings (Coating, Coating + A11) were evenly distributed on the surfaces of the experimental cheese samples by spraying until fully covered. The control acid-curd cheese (no treatment, C) and four experimental cheeses (treated with plain AWPC (C + AWPC), AWPC fermentate (C + AWPC Fermentate A11), plain coating (C + Coating), and with coating with *L. paracasei* A11 incorporated (C + Coating + A11)) were covered with plastic lids and stored refrigerated at 4 ± 1 °C for 14 days.

### 2.8. Sample Analysis

The sugar content in fermentates was measured in triplicate according to methods: liquid chromatographic determination for specific sugars [28] and an enzymatic method for lactic acid and lactate content [29].

Cheese samples were analysed in triplicate on days 1, 8, and 14 of storage for microbiological and pH changes, along with overall acceptability.

pH was measured directly with a pH meter (Sartorius Professional meter for pH Measurement, Germany).

On days 1, day 8, and day 14 of acid-curd cheese storage, viable counts of bacteria typically found in this type of acid-curd cheese were tested in triplicate on the selective media for each group of microorganisms: MRS agar, SDA, and Violet Red Bile Glucose Agar (Oxoid, Basingstoke, UK). The following methods were used to count the microorganisms: total mesophilic LAB count [30], yeast and mould count [31] and enterobacteria count [32].

Following the prior confirmation of microbiological safety, sensory acceptance was examined on days 1 and 14 of acid-curd cheese storage. A professional panel of 7 people evaluated the samples in the sensory room (ages between 20 and 50 years old, both genders). The panel had already been chosen and trained in accordance with the International Organisation for Standardisation guidelines [33]. The samples were served at room temperature and were coded with 3-digit random numbers. The panel members received samples of curd cheese at random on identical plastic plates. A 10-point hedonic scale, ranging from 1 (dislike extremely) to 10 (like extremely), was used to evaluate sensory acceptability.

### 2.9. Statistical Analysis

SPSS statistical program (SPSS 27, SPSS Inc., Armonk, NY, USA) was used for data processing and analysis. The factors of the acid-curd cheese storage period and cheese treatment were analysed using descriptive statistics (Explore) and GLM methods. A multiple comparison Tukey test was used to compare the results at a 95% confidence level.

## 3. Results and Discussion

### 3.1. Haemolysis and Antibiotic Susceptibility Evaluation of Isolated Lactobacilli

The virulence factor of microorganisms is the presence of haemolysis which can result in the epithelial layer of the intestines breaking down. This consequently results in the inevitable testing of candidate strains for this reaction, requiring the strains to produce no haemolysis zone, which are γ-haemolytic [34]. We observed no clear transparent or green zones around microorganism colonies on the blood agar plates. Thus, colonies were found γ-haemolytic except for *Lacticaseibacillus rhamnosus* L1, which indicated α-haemolysis (Table 1). Studies showed that most of the isolates from various dairy matrices were γ-haemolytic [34,35,36].

Screening for antibiotic resistance of isolated LAB strains must be performed as they pose a risk of horizontal antibiotic gene transfer to pathogenic strains. Therefore, EFSA (2012) has established certain cut-off antibiotic resistance values for different LAB species. The isolate is considered resistant when it can grow at a concentration higher than the established cut-off value [37]. In our study, the antibiotic susceptibility of the isolates was documented by using different commonly used antibiotics (Table 1); 9 out of 12 tested isolates exceeded cut-off values for a certain antibiotic. *Lacticaseibacillus rhamnosus* L1, *Lactobacillus amylovorus* PR41 and four *Lacticaseibacillus paracasei* isolates, A161-1, A173-2, R111, and R112, were resistant to chloramphenicol, while R112 were also resistant to kanamycin and PR41—to tetracycline. In addition, three *Lactiplantibacillus plantarum* isolates, PR21, PR23, and PR35, were resistant to kanamycin.

The highest level of resistance was noted by PR21, PR23 isolates to kanamycin (>256 µg mL^−1^), L1 to chloramphenicol (16 µg mL^−1^), and PR41 to tetracycline (12 µg mL^−1^). Wide spreading resistance of lactobacilli strains to chloramphenicol and kanamycin was also reported by other researchers [38,39].

All 12 tested isolates demonstrated intrinsic resistance to vancomycin (>256 µg mL^−1^). According to Klare et al., [40], intrinsic resistance to vancomycin is very common among lactobacilli.

Only three isolates out of 12—*Lacticaseibacillus paracasei* A11, *Lactiplantibacillus plantarum* A154-d1, *Lactiplantibacillus plantarum* PR33—met EFSA requirements for antibiotic resistance and were selected for further analysis.

### 3.2. Enzymatic Activity Evaluation of Lactobacilli

Some enzymes excreted by microorganisms can impact human health to various degrees. For example, beta-glucuronidase and beta-glucosidase convert aromatic hydrocarbons and amines into active carcinogens, increasing the risk of colon cancer, while β-galactosidase reduces the symptoms of lactose maldigestion [41]. Decarboxylases can produce biogenic amines, such as histamine and tyramine, resulting in allergic reactions [42]. Specific activities of hydrolytic enzymes for each isolate in our study were determined using the micro enzyme API ZYM system, and the results are shown in Table 2. None of the tested lactobacilli showed activity of alkaline phosphatase, lipase, trypsin, alpha-galactosidase, beta-glucuronidase, alpha-mannosidase and alpha-fucosidase. On the other hand, the strongest enzymatic reactions were demonstrated by all isolates of leucine arylamidase and valine arylamidase. All isolates showed enzymatic activity of the Naphthol-AS-BI-phosphohydrolase enzyme, where the strongest activity was detected in the A11 isolate. Moreover, the isolates A154-d1 and PR33 showed a strong enzymatic reaction of beta-galactosidase. PR33 also showed a strong reaction of alpha-glucosidase, beta-glucosidase, and a weak reaction of °N-Acetyl-beta-glucosaminidase, while a strong reaction of alpha-glucosidase was seen in A11 and beta-glucosidase in A154-d1 isolates. Weak enzymatic activity of esterase (C4), esterase lipase (C8), cystine arylamidase, alpha-chymotrypsin, acid phosphatase, alpha-glucosidase, °N-Acetyl-beta-glucosaminidase were detected in few of the tested isolates. There is a wide variation among indigenous strains in enzymatic activity; other studies reported beta-glucuronidase and beta-glucosidase enzymatic activity in 99 tested LAB strains [43].

Only one isolate, *L. paracasei* A11, met all safety requirements and was chosen for further experiments.

### 3.3. Inhibition of Yeasts and Moulds by L. paracasei A11

Yeasts and moulds, which frequently cause dairy products to spoil, can change the appearance of the food due to their proliferation on the surface. Different yeasts, such as *Candida species*, *Yarrowia lipolytica* [44], as well as the moulds *Cladosporium, Aspergillus, Mucor* and *Penicillium* [45,46], that are commonly reported as responsible for fresh cheese spoilage were selected for testing antifungal properties of the strain in a dual culture overlay assay (Figure 2). The experiment results showed strong (+++) *L. paracasei* A11 activity against *D. hansenii*, and *Cl. herbarum*, minimal (+) inhibitory effect of the strain on *P. commune*, *A. versicolor* and *Y. lypolitica*, and *C. glabrata*. The strain was not active against *M. racemosus* and *C. albicans*. This is in agreement with [47], reporting only seven antifungal strains out of the 56 tested, with *L. paracasei* expressing the highest antifungal activity among them. Other authors reported strong antifungal activity of the *L. paracasei* strain against *M. racemosus* [48], which we did not observe in our experiment, pointing to the strain-dependent nature of antifungal activity [49]. On the other hand, a relationship between LAB and proteins that are involved in the stress response could play a crucial role in the strain’s antifungal activity, suggesting that lactobacilli may express higher antifungal properties in real food matrices [44].

### 3.4. Growth of L. paracasei A11 in AWPC

The optical density (OD) estimated by the bioreactor and the rate of bacteria growth in MRS broth and AWPC are presented in Figure 3a,b.

The impacts of growth media, time and their interaction on OD, growth rate, and biomass yield were significant (*p* < 0.001). We observed a rapid change in OD in MRS broth; the recorded doubling time of the strain was 2.14 h in this growth medium (Figure 3a). The mean content of glucose decreased while lactic acid and lactate content increased significantly (*p* < 0.001, Table 3) in MRS broth, followed by a significant drop in pH (Figure 4b).

Cultivation in AWPC induced a diauxic growth of *L. paracasei* A11 strain in the first exponential phase (E1) with a doubling time of 3.5 h due to effective consumption of glucose (*p* < 0.001, Table 3). The doubling time in the second phase (E2) was 15 h (Figure 3b) when the strain adapted to consume galactose (*p* = 0.09) and lactose (*p* = 0.1), producing lactic acid and lactates (*p* < 0.05; Table 3). Adaptation of the strain to an acidic environment in AWPC led to a gradual decrease of pH compared to a rapid pH drop in MRS broth (Figure 4b; 4.49 and 3.46, respectively).

After 24 h of cultivation, the cfu count in the cultures grown in MRS broth was significantly higher than that in AWPC (8.46 and 6.85 log_10_ cfu mL^−1^, respectively (Figure 4a). Importantly, after 48 h of cultivation in MRS broth, the cfu count (9.23 log_10_ cfu mL^−1^) remained statistically unchanged compared to the 24-h-old culture, whereas the cfu count in AWPC cultivates increased significantly (8.3 log_10_ cfu mL^−1^) compared to the 24-h-old culture (*p* < 0.001). So far, few publications have provided information on the use of acid or lactic whey in the preparation of LAB culture medium. Dudkiewicz et al. [20] investigated the filtration and sterilisation of acid whey to cultivate different yeast strains. There was also an attempt to grow LAB in an acid whey-based medium supplemented with low levels of yeast extract. The authors reported that LAB in an acid whey-based medium supplemented with 30% tomato juice and 1% yeast extract produced similar amounts of biomass as that obtained from MRS medium [21], despite the fact that the cultivation of strain in AWPC yielded less (*p* < 0.01) compared to that in MRS broth, the fermentate produced from growth in AWPC was suitable for other applications since no additional supplementation to the medium was used.

### 3.5. Evaluation of Antifungal Activity of L. paracasei A11 Fermentate and Coatings on Acid-Curd Cheese

To test the protective properties of AWPC fermentates and their coatings in situ, they were deposited on the surface of fresh acid-curd cheese. The single storage and treatment factors and their interactions had a significant impact (*p* < 0.001) on the pH and microbiological parameters of acid-curd cheese (Figure 5a–e). Curd cheese is known for its whitish colour, soft, grainy texture, and acidic flavour mainly due to the metabolic activity of starter and non-starter LAB [50]. In this study, pH increased on day eight and then decreased only in control cheese due to natural protein hydrolysis observed in our previous experiment with acid-curd cheese [51] (Figure 5a). All treated samples demonstrated a decrease in pH on day eight and then an increase on day 14. We speculate that the natural cheese proteolysis was slowed down in experimental cheese samples due to the treatment with AWPC. No statistical differences in pH among samples we found on Day 14.

LAB counts increased steadily in all samples during 14 days of storage (Figure 5b). LAB growth, though in the samples with *L. paracasei* A11 strain, was less pronounced (*p* < 0.001) during the whole storage period due to the antagonistic activity of the strain, causing suppression of protein breakdown and, thus, maintaining cheese freshness (Figure 5a,f). This is in agreement with Mileriene et al. [17] reporting lower counts of LAB and the lower mean content of water-soluble nitrogen hermoso-coagulated acid-whey protein cheese supplemented with indigenous *Lactococcus lactis* during eight days of storage that resulted in a considerable increase in overall acceptability.

Fresh cheese is a good target for mould and yeast spoilage, resulting in a short (7–8 days) shelf life [1,52]. In our study, we observed rapid growth of fungi in all samples not supplemented with *L. paracasei* A11, reaching 7.4–7.6 log_10_ cfu mL^−1^ at the end of storage. Plain coating and coating with the *L. paracasei* A11 strain significantly lowered the yeast counts at day one (*p* < 0.05) (Figure 5c). Plain coating, as well as AWPC fermentate and coating with the *L. paracasei* A11 strain significantly reduced the yeast counts on day eight (*p* < 0.001). Only the *L. paracasei* A11 supplemented coating was able to suppress the yeast counts for 1 log_10_ cfu mL^−1^ on day 14 (*p* < 0.001). Similar dynamics were observed in mould counts: while plain coating and fermentate significantly reduced mould counts on day eight, the coating with *L. paracasei* A11 was able to suppress mould growth throughout the storage (reduction of 1.0, 2.0, 1.5 log_10_ cfu mL^−1^ compared to control on day one, eight, and 14, respectively) (Figure 5d).

We also detected significantly less *Enterobacteriaceae* in all experimental samples during the storage compared to the control ones (Figure 4e). The supplementation with *L. paracasei* A11 proved its antibacterial properties reducing enterobacteria counts in the samples covered with its fermentate (*p* < 0.05), and totally suppressed their growth from day eight in the samples covered with its coating (*p* < 0.001). In agreement with our study, Aunsbjerg et al. [53] reported antifungal activity of a single *L. paracasei* strain in yoghurt. Other study proved the antifungal activity of mixed cultures of *L. paracasei* subsp. *paracasei* SM20, SM29, or SM63 and *Propionibacterium jensenii* SM11 applied on the surface of cheese [54].

Microbial contamination is frequently the cause of sensory issues in fresh curd cheese [44]. As a result, bioprotective LAB cultures are utilised to stop product spoiling and maintain flavor. LAB cultures, especially if used in a form of fermentates containing antifungal metabolites, may have an influence on sensory acceptance and noticed by the consumer. Therefore, the sensory acceptance of acid-curd cheese in our experiment was evaluated in the beginning (day 1) and in the end (Day 14, Figure 5f) of the experiment. Treatment did not influence cheese acceptance (*p* > 0.05) while storage time and their interaction influenced it significantly (*p* = 0.05). On the contrary, sweet whey protein concentrate based coating that was used on the same type of cheese by Mileriene et al. [1] did not influence the flavor of cheese. Plain AWPC coating in our study due to its sour taste and viscous texture received lower scores compared to other samples on day 1 (*p* < 0.05) and was described by the panelists as “acidic”. Supplementation of the coating with *L.paracasei* A11 had a positive effect on cheese flavor that was noted as “balanced” on day 1. In the end of our experiment, both coated samples, as well as AWPC fermentate demonstrated higher sensory acceptance (*p* < 0.05) compared to control sample and samples sprayed with AWPC alone (C + AWPC). The development of spoilage microorganisms during storage in control (CC) cheese and cheese sprayed with AWPC (Figure 5c,d) decreased scores of the acceptability criteria due to various off-odors and off-flavors.

## 4. Conclusions

Growing demand for natural food with less chemical preservatives calls for isolation and screening of various indigenous protective LAB cultures to be employed in reducing the spoilage of perishable dairy products, such as fresh cheese. Despite the fact that LAB is generally recognised as safe, each new candidate strain has to be not only evaluated for their protective properties but also for their safety. We found that one isolate *L. paracasei* A11 out of 12 after screening met all safety requirements and expressed strong broad-spectrum antifungal activity. Cultivation of that strain in AWPC produced similar amounts of biomass as that obtained from the MRS medium. The fact that some acid-tolerant LAB species can effectively reproduce in plain acidic dairy by-products opens the opportunity to employ this growth medium for commercial biomass production. Further studies are needed to test whether supplementation of AWPC could contribute to more rapid growth and higher biomass yield. Utilisation of such AWPC fermentate as a vehicle of protective strain in preparation of AWPC-pectin based edible coating formulation proved to be a valid, innovative method, allowing repurposing acid whey, saving on expensive and non-food grade LAB growth media, such as MRS broth, and avoiding biomass preparation procedure costs (centrifugation etc.). It could be beneficial for cheese producers to reintroduce AWPC, an acid-curd cheese by-product, as a foundation for edible coating. After the manufacturing of acid-curd cheese, this coating might be made from leftover acid whey. This type of bioactive antimicrobial coating could be of interested for cheese manufacturers aiming for sustainability, enhanced quality and extended shelf-life of the final product.

## Figures and Tables

**Figure 1 microorganisms-11-00557-f001:**
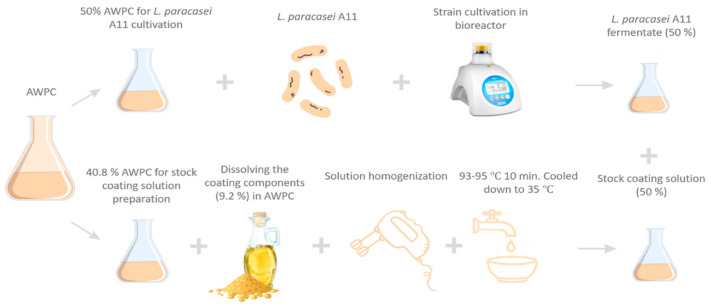
Scheme of the process followed for preparation of coatings.

**Figure 2 microorganisms-11-00557-f002:**
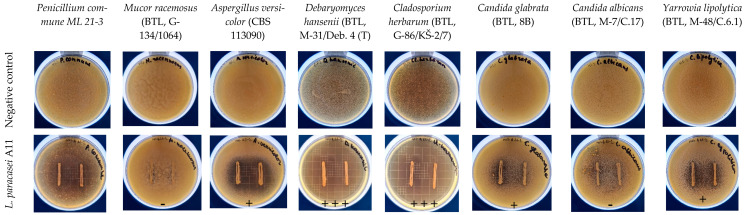
Activity of *L. paracasei* A11 isolate against four different moulds and four different yeasts after 48 h of aerobic incubation at 25 °C. Negative control shows petri plates with moulds/yeasts and without lactobacilli. The following scale was used: +++, no fungal growth on 8% of plate area/bacterial streak; +, no fungal growth on 0.1–3% of plate area/bacterial streak; -, no visible inhibition. Inhibition tests were performed in triplicate.

**Figure 3 microorganisms-11-00557-f003:**
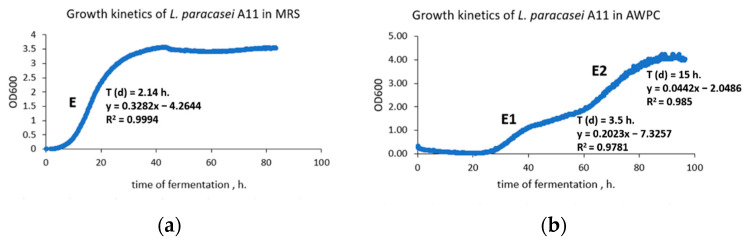
Growth kinetics of *L. paracasei* A11 in MRS broth (**a**) and acid whey protein concentrate (AWPC, (**b**)). X-axis: time (h) of cultivation. Y axis: Optical density (OD). Growth phases: exponential phase ©, first exponential phase (E1) and second exponential phase (E2).

**Figure 4 microorganisms-11-00557-f004:**
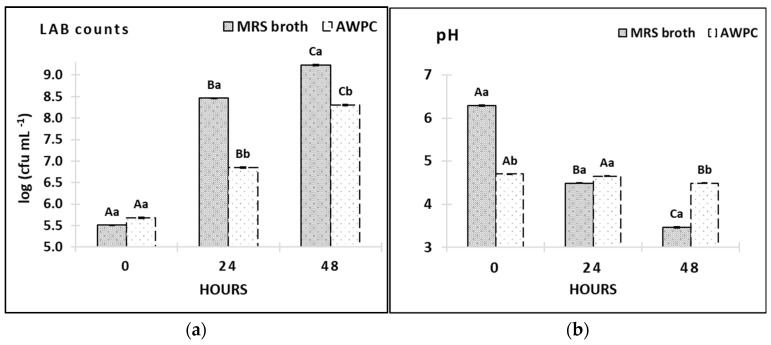
LAB counts (**a**) and pH (**b**) in MRS broth and AWPC *L. paracasei* A11 fermentates (AWPC) kept in a culture flask for 48 h at 37 °C. Means among storage days within the same cheese treatment marked with different upper-case letters are significantly different (*p* ≤ 0.05). Means between cheese treatments within the same storage day marked with different lower-case letters are significantly different (*p* ≤ 0.05).

**Figure 5 microorganisms-11-00557-f005:**
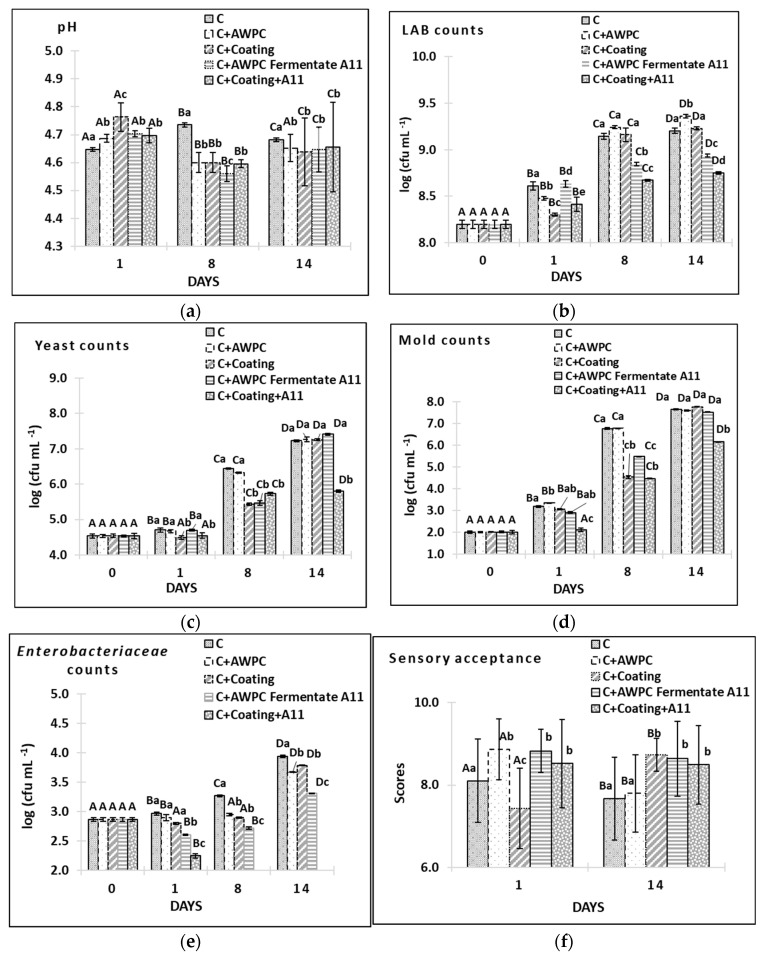
pH (**a**), LAB counts (**b**), yeast counts (**c**), mould counts (**d**), *Enterobacteriaceae* coun© (**e**) and sensory acceptance (**f**) in control acid-curd cheese (C), acid-curd cheese coated with plain AWPC (C + AWPC), acid-curd cheese coated with plain coating (C + Coating), acid-curd cheese coated with AWPC *L. paracasei* A11 fermentate (C + AWPC Fermentate A11) and acid-curd cheese coated with coating with *L. paracasei* A11 fermentate (C + Coating + A11) during 14 days of storage at 4–6 °C. Means among storage days within the same cheese treatment marked with different upper-case letters are significantly different (*p* ≤ 0.05). Means among cheese treatments within the same storage day marked with different lower-case letters (a–d) are significantly different (*p* ≤ 0.05).

**Table 1 microorganisms-11-00557-t001:** Haemolysis and antibiotic susceptibility of lactobacilli isolates.

Isolate	Identification	Haemolysis	Ampicillin	Erythromycin	Clindamycin	Chloramphenicol	Streptomycin	Gentamycin	Kanamycin	Vancomycin	Tetracycline
A161-1	*L. paracasei*	γ	1 (4)	0.19 (1)	0.50 (1)	6 (4)	24 (64)	4 (32)	64 (64)	>256 (n.r.)	0.5 (4)
A11	*L. paracasei*	γ	0.75 (4)	0.19 (1)	0.64 (1)	4 (4)	12 (64)	6 (32)	24 (64)	>256 (n.r.)	0.5 (4)
A173-2	*L. paracasei*	γ	0.75 (4)	0.25 (1)	0.75 (1)	8 (4)	24 (64)	6 (32)	48 (64)	>256 (n.r.)	0.75 (4)
A154-d1	*L. plantarum*	γ	0.125 (2)	0.25 (1)	0.75 (2)	6 (8)	8 (n.r.)	3 (16)	12 (64)	>256 (n.r.)	4 (32)
PR33	*L. plantarum*	γ	0.032 (2)	0.5 (1)	<0.016 (2)	2 (8)	32 (n.r.)	4 (16)	64 (64)	>256 (n.r.)	6 (32)
L1	*L. rhamnosus*	α	0.75 (4)	0.94 (1)	0.75 (1)	16 (4)	4 (32)	1.5 (16)	8 (64)	>256 (n.r.)	0.75 (8)
PR21	*L. plantarum*	γ	0.032 (2)	0.19 (1)	0.047 (2)	2 (8)	64 (n.r.)	12 (16)	>256 (64)	>256 (n.r.)	2 (32)
PR23	*L. plantarum*	γ	0.047 (2)	0.38 (1)	0.047 (2)	4 (8)	64 (n.r.)	12 (16)	>256 (64)	>256 (n.r.)	6 (32)
PR35	*L. plantarum*	γ	0.047 (2)	0.38 (1)	0.016 (2)	4 (8)	48 (n.r.)	4 (16)	96 (64)	>256 (n.r.)	6 (32)
PR41	*L. amylovorus*	γ	0.19 (1)	0.38 (1)	0.38 (1)	8 (4)	6 (16)	2 (16)	12 (16)	>256 (2)	12 (4)
R111	*L. paracasei*	γ	0.094 (4)	<0.016 (1)	0.47 (1)	8 (4)	16 (64)	3 (32)	32 (64)	>256 (n.r.)	1,5 (4)
R112	*L. paracasei*	γ	<0.016 (4)	<0.016 (1)	0.19 (1)	8 (4)	8 (64)	4 (32)	192 (64)	>256 (n.r.)	0.75 (4)

n.r. not required. Cut-off values that are indicated in parentheses are provided by EFSA.

**Table 2 microorganisms-11-00557-t002:** Enzymatic activity of selected isolates.

Enzyme	Lactobacilli Isolates		
	A11	A154-d1	PR33
Alkaline phosphatase	0	0	0
Esterase (C4)	2	0	0
Esterase lipase (C8)	1	0	0
Lipase (C14)	0	0	0
Leucine arylamidase	5	4	4
Valine arylamidase	5	3	4
Cystine arylamidase	1	0	1
Trypsin	0	0	0
Alpha-chymotrypsin	1	0	0
Acid phosphatase	2	0	0
Naphthol-AS-BI phosphohydrolase	3	2	2
Alpha-galactosidase	0	0	0
Beta-galactosidase	0	3	3
Beta-glucuronidase	0	0	0
Alpha-glucosidase	4	0	3
Beta-glucosidase	0	4	4
N-Acetyl-beta-glucosaminidase	0	4	2
Alpha-mannosidase	0	0	0
Alpha-fucosidase	0	0	0

Colour reaction grade 0 on the API-ZYM test scale was interpreted to correspond to a negative reaction, grades 1 and 2 corresponded to a weak reaction, and grades 3, 4 and 5 corresponded to a strong reaction.

**Table 3 microorganisms-11-00557-t003:** Time course of lactic acid and lactate production and reducing sugars consumption of selected *L. paracasei* A11 in MRS broth and acid whey protein concentrate (AWPC).

Parameter	Growth Media			
	MRS Broth		AWPC	
	0 h	48 h	0 h	48 h
**Glucose (%)**	1.83 ± 0.19 ***^a^	0.28 ± 0.12 *^a^	0.66 ± 0.01 ***^b^	0.57 ± 0.03 *^b^
**Galactose (%)**	-	-	1.05 ± 0.04 ***	0.97 ± 0.03 ***
**Lactose (%)**	-	-	9.84 ± 0.06 ***	9.75 ± 0.09 ***
**D/L lactic acid (mg/100 g)**	34.45 ± 0.46 ***^a^	1481.85 ± 51.99 *^a^	847.03 ± 1.65 ***^b^	1030.70 ± 58.85 *^b^
**D/L lactate (mg/100 g)**	34.06 ± 0.45 ***^a^	1465.50 ± 51.41 *^a^	837.62 ± 1.63 ***^b^	1019.26 ± 58.19 *^b^

Values presented are means of three replicates ± standard deviation. Means in the same row with different lowercase letters indicate significant differences (*p* < 0.05) between treatments within the same time. Means between time points within the same treatment are statistically different when *p* < 0.05 (*), and *p* < 0.001 (***).

## Data Availability

Data sharing not applicable.

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
