# Peer review of "Screening for Antifungal Indigenous Lactobacilli Strains Isolated from Local Fermented Milk for Developing Bioprotective Fermentates and Coatings Based on Acid Whey Protein Concentrate for Fresh Cheese Quality Maintenance"

_microorganisms, 2023, doi:10.3390/microorganisms11030557_

Round 1

Reviewer 1 Report

In the present work, L. paracasei A11, an indigenous lactobacilli isolated from cow milk, was selected for their safety and antifungal activity. This strain was used for the development of bioprotective AWPC fermentates and coatings, evaluating their potential use for fresh cheese quality maintenance. Overall, the work is interesting and the results are meaningful. However, some details must be modified and correct before publication.

-Punctuation marks must be revised along the manuscript. In particular, several commas are missing which makes difficult the fluid reading of the manuscript.

-Tittle: “developing of” should be replace by “developing” or “development of”.

-Abstract: Please indicate the meaning of AWPC the first time is mentioned in the abstract (Page 1-line 24).

-Page 2

Line 1: Indicate the meaning of EU

-Page 3

 Lines 1-3 and Lines 15-17: These paragraphs are describing results so I suggest moving them to the corresponding section (3. Results and discussion).

-Page 4

Line 17:  2.4.4. Antifungal activity assays of L. paracasei A11. The evaluation of antifungal activity do not correspond to a safety characterization of the strain. Thus, it should not be included in the Section 2.4 Safety assessment. It should be in a complete separate section.

Line 23: “two 2 cm” Please, correct.

Lines 36-37: In the sentence “The L. paracasei A11 strain…” remove the word “the”. In the sentence “…expressed strong broad-spectrum antifungal activity were chosen” replace “were” with “was”.

Line 40: what do the authors mean with “the physiological activity”? Please, specify.

-Page 5

Line 35: I suggest mentioning the methods used to determine sugars, lactic acid and lactate.

-Page 6

Line 1: “selective media for each species”. I think it would be more appropriate to say “selective media for each group of microorganisms”

Line 2: specify the medium for each group of microorganisms.

Line 21: The virulence factor is the presence of hemolysins.

Line 23: “.. produce no haemolysis zone”.

Line 43-48: The authors mentioned that “Lacticaseibacillus paracasei A11, Lactiplantibacillus plantarum A154-d1, Lactiplantibacillus plantarum PR33 were found susceptible to most of tested antibiotics, except vancomycin.” However, this resistance to vancomycin is an intrinsic resistance very common among lactobacilli. What is the relevance of this resistance? Are the strains safe because this resistance is intrinsic? This should be clarify in the text.

-Table 1: Please correct the rows of PR33 and PR35, which are not aligned. Correct the MIC of R111 against Clindamycin. In the footnote replace “Values above the breakpoint” with “Cut off values”

-Page 8

3.2. Enzymatic activity evaluation of lactobacilli. As mentioned by the authors, beta-glucuronidase and beta-glucosidase are enzymes with negative impact in consumers’ health. Is there any other enzyme that should be absent in a safe strain? If so, please add it in the text.

Line 13: Replace “expression” with “activity”

-Page 9

Line 12: Replace “to” with “against”

-Page 11

Figure 2: The legend do not correspond to the figure. Please, correct.

-Page 12

Figure 3: “Means among storage days within the same cheese treatment marked with different upper-case letters are significantly different (p ≤ 0.05). Means between cheese treatments within the same storage day marked with different lower-case letters (A-C) are significantly different (p ≤ 0.05).” This do not correspond to the Figure. Please, correct.

-Page 13

Line 16: “There was no difference in yeast counts on day 1 in all samples”. However, Figure 5c shows that  coating and coating + A11 presented significantly lower counts of yeast at day 1.

-Page 14

Figure 5 is actually Figure 4.

Enterobacteriaceae counts are showed in the Figure but they are not discussed in the text. Please include the description of these results in the manuscript.

Author Response

We are grateful for the opportunity to improve our manuscript. Below please find the answers to all reviewers’ questions. All changes in the manuscript have been marked up using the “Track Changes” function.

Reviewer 2 Report

This study investigates the effectiveness of a strain of L. paracasei as biocontrol agent to increase the safety and shelf life of fresh cheese. The study is complete and it encompass several steps including i) isolation; ii) in vitro determination of the safety; ii) in vitro determination of the antifungal potential; iv) validation of AWPC as alternative and cheap substrate to obtain microbial biomass; v) application in cheese by comparing different strategies of inoculum (i.e. with or without coating).

Although several LAB strains have been proposed as bioprotective cultures in cheese-making, the employment of AWPC as substrate to obtain value microbial biomass is the main novelty of the study.

However, some critical concerns should be addressed.

The authors selected strain A11 after its safety evaluation and due to its broad-spectrum antagonism against filamentous fungi and yeast that are typical spoilage of cheeses. In contrast, in vivo assays have been performed to assess the bio-protective potential of strain A11 against spontaneous microbial contamination including yeast, moulds, lactic acid bacteria, and Enterobacteriaceae. In my opinion, artificially contaminated samples with the same strains employed for the in vitro assay should be tested. Moreover, the antibacterial activity is not corroborated by any in vitro evidence.

I don’t understand figure 5 b. Samples inoculated with strain A11 at least at time zero should show a load of LAB higher, since they have been inoculated with 7.7 log10 cfu/mL. If differences have been not detected at this experimental time, probably all the results are questionable. To better interpret the results, monitoring of viability of A11 should be performed. Moreover, a clear explanation of the how the experiment was done should be provided in materials and methods.

In particular:

L28-P5. samples by spraying until full coverage. how much volume was used? the same for all samples?

L40-P5. How was the microbiological analysis carried out? what amount of sample was used? was the rind or the whole cheese analysed?

L3-P3. Only 12 strains out of 53 were selected for further analyses due to strict purification procedures. what does it mean? you just said you isolated 126 strains of which 53 were lactobacilli. If they were not pure cultures, you cannot report these numbers as isolates or strains.

Presumptive LAB have been isolated and grown under anaerobic conditions. However, to estimate the yield biomass in AWPC strain A11 has been grown under aerobic conditions.

Author Response

We are grateful for the opportunity to improve our manuscript. Below please find the answers to all reviewers’ questions. All changes in the manuscript have been marked up using the “Track Changes” function.

Please see the answers in the attachment.

Reviewer 3 Report

The topic of the manuscript: „Screening for antifungal indigenous lactobacilli strains isolated from local fermented milk for developing of bioprotective fermentates and coatings based on acid whey protein concentrate for fresh cheese quality maintenance falls within the thematic scope of the MICROORGANISMS.

The aim of the study was to screen the lactobacilli isolated from fermented milk to select the strain with strongest fungicidal properties for the development of bioprotective acid whey protein concentrate based fermentates and coatings intended for fresh cheese quality maintenance.

The main remarks concerns to:

1. “Abstract” and “Introduction” – require minor adjustments (line 24, 30 and 31/ page 1; line 4, 7, 38/page 2),

2. Subchapter 2.4. title “Safety assessment” - not sure what the safety is about,

3. “Materials and methods” - I would suggest in diagram 1 to add captions, what is in the flasks and what is happening at each stage (+ change Scheme 1 to Figure 1),

4. “Results”: on page 11 Figure 2 - Wrong Figure 2, wrong title of the drawing, but I don't know if the course of the curves as well. There is no correlation between the drawing and its description in the text (page 11, line 10).

5. References - numerous gaps in bibliographic data.

All suggestions for corrections (not only mentioned above) were introduced in the review mode to the attached pdf file.

Author Response

(The authors gave the same response as above.)

Round 2

Reviewer 2 Report

The revised version of the manuscript has been improved and it is now suitable for publication in Microorganisms